# Challenges in Diagnosing Focal Liver Lesions Using Contrast-Enhanced Ultrasound

**DOI:** 10.3390/diagnostics15010046

**Published:** 2024-12-28

**Authors:** Tudor Voicu Moga, Raluca Lupusoru, Mirela Danila, Ana Maria Ghiuchici, Alina Popescu, Bogdan Miutescu, Iulia Ratiu, Calin Burciu, Teofana Bizerea-Moga, Anca Voron, Ioan Sporea, Roxana Sirli

**Affiliations:** 1Department of Gastroenterology and Hepatology, “Victor Babeș” University of Medicine and Pharmacy, 300041 Timișoara, Romania; moga.tudor@umft.ro (T.V.M.); raluca.lupusoru@umft.ro (R.L.); ghiuchici.anamaria@umft.ro (A.M.G.); isporea@umft.ro (I.S.);; 2Center of Advanced Research in Gastroenterology and Hepatology, “Victor Babeș” University of Medicine and Pharmacy, 300041 Timisoara, Romaniaclaudia.voron@gmail.com (A.V.); 3Department of Gastroenterology, Faculty of Medicine, Pharmacy and Dental Medicine, “Vasile Goldis” West University of Arad, 310414 Arad, Romania; 4Department of Pediatrics-1st Pediatric Discipline, “Victor Babeș” University of Medicine and Pharmacy, 300041 Timisoara, Romania; bizerea.teofana@umft.ro

**Keywords:** focal liver lesions, contrast-enhanced ultrasound (CEUS), hepatocellular carcinoma (HCC) diagnosis

## Abstract

**Background:** Contrast-enhanced ultrasound (CEUS) has become the preferred method for many clinicians in evaluating focal liver lesions (FLLs) initially identified through standard ultrasound. However, in clinical practice, certain lesions may deviate from the typical enhancement patterns outlined in EFSUMB guidelines. **Methods:** This study aims to assess FLLs that remained inconclusive or misdiagnosed after CEUS evaluation, spanning eight years of single-center experience. Following CEUS, all FLLs underwent secondary imaging (CT, MRI) or histopathological analysis for diagnostic confirmation. **Results:** From the initial 979 FLLs, 350 lesions (35.7%) were either inconclusive or misdiagnosed by CEUS, with hepatocellular carcinoma (HCC) and liver metastases constituting the majority of these cases. The most frequent enhancement pattern in inconclusive lesions at CEUS was hyper-iso-iso. Factors such as advanced liver fibrosis, adenomas, and cholangiocarcinoma were significantly associated with higher rates of diagnostic inaccuracies. **Conclusions:** Advanced liver fibrosis, adenomas, and cholangiocarcinoma were significantly associated with increased diagnostic challenges, emphasizing the need for supplementary imaging techniques.

## 1. Introduction

In daily practice, focal liver lesions are commonly encountered, among which hepatocellular carcinoma (HCC) stands out as the most prevalent type of primary liver cancer, affecting over 500,000 people worldwide annually [1]. Although HCC is the most frequent lesion depicted among patients with liver cirrhosis [2], other focal liver lesions can also manifest in cirrhotic and noncirrhotic livers, occurring with a prevalence of approximately 20% [3].

Contrast-enhanced ultrasound (CEUS) has become the method of choice for many practitioners for the evaluation of focal liver lesions (FLL) detected by standard ultrasound and serves as the initial imaging choice for examining FLLs in many centers due to its performance and cost-effectiveness [4,5,6,7,8,9]. CEUS offers distinct advantages over CT and MRI, including the ability to monitor contrast flow in real-time, which is essential for detecting arterial phase hyperenhancement without the timing issues related to patient variability. Ultrasound contrast agents are strictly intravascular; thus, CEUS directly visualizes blood flow and lesion perfusion, as opposed to the potential extravascular contrast spread seen in CT/MRI [10]. Importantly, CEUS is safe for patients with renal impairment due to non-nephrotoxic microbubbles, allows for multiple injections for a thorough examination, and offers the flexibility to clear and reevaluate the contrast in the bloodstream with adjusted settings for detailed arterial phase analysis. Microbubbles utilized in CEUS can be administered multiple times for enhanced evaluation if initial results are unclear or when various liver conditions need to be explored, thanks to their rapid clearance from blood vessels. Moreover, adjusting the mechanical index to higher settings swiftly removes microbubbles from the viewing area, permitting the detailed and repeated analysis of the arterial phase with just one dose [8,11].

However, in a real-world setting, some lesions may not follow the typical enhancement pattern described by EFSUMB guidelines [8,12]. Previous studies have highlighted concerns regarding CEUS potentially leading to the incorrect identification of intrahepatic cholangiocarcinoma (ICC) as HCC in cirrhotic livers [13]. Another review [14] emphasized the complexity of diagnosing benign liver lesions, which are commonly detected incidentally via imaging. It stresses the importance of sophisticated diagnostic tools such as MRI with hepatobiliary contrast or CEUS to differentiate these benign lesions, including focal nodular hyperplasia (FNH), hepatocellular adenoma (HCA), and hemangiomas, from malignant ones, highlighting the diagnostic challenges posed by atypical presentations and the similarity of enhancement patterns between some benign and malignant lesions. Understanding the various benign and malignant lesions that may pose diagnostic challenges on B-mode and contrast-enhanced ultrasound (CEUS), in cirrhotic and noncirrhotic patients, is crucial to prevent misdiagnosis and, consequently, inappropriate therapeutic choices. This study aimed to evaluate FLLs with ambiguous CEUS results over an eight-year span, scrutinizing the factors contributing to diagnostic inaccuracies within our patient cohort.

## 2. Materials and Methods

This study employed a retrospective, single-center design, analyzing patient data from a tertiary hepatology center in Romania. It received ethical approval from our University ethics committee according to local and national law.

### 2.1. Subjects

The analysis focused on patients admitted to our department, either as inpatients or outpatients, who had one or multiple FLLs identified through standard B-mode ultrasound evaluation and for whom gray-scale B-mode evaluation alone was insufficient for a definitive diagnosis. The indication for CEUS in our daily practice is routine for all newly detected FLLs for whom B-mode evaluation is not diagnostic. All patients were subsequently investigated using a second-line imaging method (contrast-enhanced CT, MRI, or biopsy) to confirm the diagnosis. Liver fibrosis stages were assessed non-invasively using vibration-controlled transient elastography (VCTE) performed with the FibroScan^®^ device (Echosens, Paris, France) and the 2D shear wave elastography (2D-SWE) technique available on the LOGIQ E9 system from General Electric.

A total of 979 consecutive FLLs were included in the study, evaluated in our department during an eight-year period (from January 2009 to December 2016). The main focus was on the lesions that were inconclusive or misdiagnosed during CEUS, as a first-line imaging method. All FLLs were judged according to EFSUMB-CEUS guidelines [8,12,15,16] and had a second-line imaging method (CT, MRI) or histology as a reference method. The time interval between the initial CEUS examination and the reference method was typically 1 to 2 days, usually within the same hospital admission. The exception was core biopsy, where the time interval for obtaining the final results ranged from 2 to 3 weeks. Using the reference method for the final diagnosis, we identified the most frequent lesions that were misdiagnosed or inconclusive, the deceiving enhancement pattern at CEUS, and reported the most frequent challenges for the examiner.

### 2.2. Exclusion Criteria

Patients with FLLs for whom gray-scale B-mode evaluation was sufficient for a final diagnosis (simple cysts, hemangioma, focal fatty sparing), or with previous known FLL, lesions without histology, or a diagnostic second-line imaging method (not fulfilling the CT/MRI diagnostic criteria for a specific lesion) were excluded from the study.

### 2.3. Contrast-Enhanced Ultrasound

For detecting lesions in gray-scale B-mode imaging, the liver was scanned with a convex array transducer, evaluating lesions by size, echogenicity, and homogeneity. When multiple FLLs were present, the study utilized the reference lesion from each CEUS examination. Two pieces of equipment from different manufacturers (Siemens Acuson S2000 (Berlin, Germany) and Logiq E9 from General Electric (Boston, MA, USA)) were utilized for CEUS investigations with conventional abdominal transducers (frequency range of 1–6 MHz). To prevent premature destruction of microbubbles, a low-mechanical-index technique (less than 0.1) was employed. The contrast agent used (SonoVue^®^, Bracco Imaging, Milan, Italy) in doses of 2.4 mL was administered as a bolus, followed by a 10 mL saline flush. Cine loops were recorded according to the EFSUMB protocol [16] during the three vascular phases: arterial (10–30 s after injection), portal (30–120 s) and late phase (>120 s). CEUS examinations were performed and interpreted by ultrasound experts (EFSUMB level 3), each with more than 5 years’ experience in CEUS imaging.

All lesions evaluated by CEUS were classified following the Guidelines and Good Clinical Practice Recommendations for CEUS in the liver [12,16]. Sustained enhancement during the portal and late phases is characteristic of nearly all solid benign liver lesions. Further characterization of these lesions can be achieved by examining their enhancement patterns in the arterial phase, such as rapid, uniform enhancement across the entire lesion (typical of focal nodular hyperplasia) or initial peripheral globular/nodular enhancement with a centripetal filling (characteristic of hemangiomas). For malignant lesions, hypo-enhancement in the late phases corresponds to the washout phenomenon characterizing malignancies. Although nearly all malignant lesions exhibit this trait, irrespective of their arterial phase enhancement pattern, there have been a few exceptions, primarily in well-differentiated hepatocellular carcinoma (HCC) [6]. The determination of an inconclusive lesion was made individually by each ultrasound expert. If there was uncertainty or disagreement, a second opinion was sought from another expert to ensure accuracy in the final CEUS assessment. Some of the lesions, due to the washout phenomenon, were considered malignant; however, they were still labeled as inconclusive due to the off-pattern enhancement in the arterial and portal phases. (Figure 1, Figure 2, Figure 3 and Figure 4).

Based on the CEUS evaluation, the following categories were defined:Correctly diagnosed by CEUS (the CEUS diagnosis matched the reference method).Misdiagnosed (the CEUS diagnosis did not align with the reference method).Inconclusive at CEUS.
C-1.Inconclusive for a definitive diagnosis but suggestive of a benign or malignant nature (the examiner could not determine the lesion type with confidence but was confident about its malignancy status).C-2.Fully inconclusive (the examiner was unable to reach any conclusion regarding the lesion’s type or malignancy).

### 2.4. Contrast-Enhanced Computer Tomography

The CT protocol for focal liver lesions involved a multi-phase contrast-enhanced approach. It included a non-contrast phase for baseline imaging, followed by sequential contrast-enhanced phases: arterial (30–35 s post-injection), portal venous (60–70 s), and delayed parenchymal (3–5 min).

### 2.5. Magnetic Resonance Imaging

The MRI protocol for focal liver lesions utilized a multi-sequence approach with contrast enhancement for optimizing lesion detection and characterization. It typically included T1-weighted and T2-weighted sequences to assess the lesion’s signal characteristics. Following this, contrast-enhanced phases are performed using gadolinium-based contrast agents. These phases encompass an arterial phase (about 20–30 s post-injection), a portal venous phase (about 60 s), and a delayed phase (up to 5 min).

### 2.6. Histology

Core needle biopsy was used as the primary technique for histological evaluation of FLLs. The procedure was guided by ultrasound to ensure accuracy in sampling and to target specific lesions identified as indeterminate or suspicious during imaging evaluations [1]. A freehand technique was employed, offering the flexibility to adjust needle placement in real-time based on lesion morphology and surrounding structures [17]. Histopathological results in this study were available for 43 patients, accounting for approximately 4.4% of the cohort. Among these cases, the predominant diagnoses were liver abscesses, HCC, and liver metastases.

### 2.7. Statistical Analysis

The statistical analysis was performed with MedCalc v19. Quantitative variables were calculated as mean ± standard deviation, while qualitative variables were calculated as percentages. Student’s t-test was used for mean comparison. Logistic regression analysis was used to identify factors implicated in the misdiagnosis of lesions with CEUS. 

## 3. Results

Out of the 979 lesions evaluated by CEUS, 629 lesions were conclusive and 350 lesions were inconclusive or misdiagnosed at CEUS (compared to the reference method) as shown in the study flowchart (Figure 5).

The main characteristics of the study sample are described in Table 1.

The mean age of the enrolled patients was 59.5 ± 12.8 years. Out of the misdiagnosed lesions (B), which were 123 out of 979, the predominant lesions were 39 HCC, 30 metastases, 22 hemangiomas, and 11 were FNH. Out of the 227 inconclusive lesions (C), 115 were classified as benign or malignant (C-1), and the rest of the 112 lesions (C-2) were considered fully inconclusive at CEUS. The most frequent inconclusive lesions (227/350) were 84 HCC, 31 metastasis, 27 hemangiomas, 12 dysplastic nodules, 11 cholangiocarcinoma, nine liver abscesses, and seven FNH. Out of the 350 FLLs (misdiagnosed or inconclusive), 42.8% were diagnosed in patients with advanced liver fibrosis, compared to only 29.4% in the CEUS conclusive group with advanced fibrosis, *p* < 0.0001.

The most frequent enhancement pattern (arterial, portal, and late phase) during CEUS of the inconclusive lesions was 61 hyper-iso-iso, 39 hyper-hypo-hypo, 22 hyper-hyper-hyper, and 21 hypo-hypo-hypo. A total of 21 lesions could not be interpreted and the rest of the enhancement pattern models were less than 10 lesions; thus, they are not mentioned.

When analyzing the misdiagnosed or inconclusive lesions in our cohort, we observed that the presence of adenomas increased the chances of incorrect diagnoses by 4.5; the presence of other neoplasms increased the chance by 1.5 times; advanced liver fibrosis by 1.8 times; and cholangiocarcinoma by 5.2 times (Table 2). The presence of hemangiomas, metastases, or female gender did not interfere with the final diagnosis. If an odds ratio is less than 1, it can be inferred that these factors may help reduce the likelihood of misdiagnoses or inconclusive results. 

When split into two groups—misdiagnosed (B) and inconclusive (C)—the same factors remain associated, meaning there are no differences between the groups. The lesions’ size and the patients’ age did not influence the correct CEUS diagnosis (Table 3 and Table 4). Regarding the presence of HCC, the findings suggest an inconsistency. There is a higher likelihood of obtaining a result with an inconclusive lesion (OR = 1.61, *p* = 0.03) compared to a misdiagnosed lesion, which is significantly less likely (OR = 0.22, *p* = 0.0001) (Table 3 and Table 4).

In a multivariate logistic regression analysis, the model included all the associated factors from the univariate analysis for misdiagnosed or inconclusive CEUS results; advanced liver fibrosis (*p* = 0.04), other neoplasms (*p* = 0.04), and adenomas (*p* = 0.0005) were independent factors associated with the misdiagnosed or inconclusive CEUS diagnostics. (Table 5).

Starting with the factors associated with misdiagnosed or inconclusive lesions in a multivariate logistic regression analysis, we tried some logistic regression models to see what happens with other related variables. The results are as follows:
Model 1: Advanced liver fibrosis and HCC; OR for misdiagnosing HCC in the presence of advanced liver fibrosis was 0.47, with *p* < 0.0001, indicating that HCC is easier to diagnose when advanced liver fibrosis is present.Model 2: Other neoplasms + advanced liver fibrosis + HCC; for an inconclusive diagnosis of HCC; OR = 1.50, *p* = 0.03, meaning that the presence of other neoplasms can raise problems in HCC diagnostics.Model 3: Advanced liver fibrosis + metastasis; OR = 0.93, *p* = 0.0004, indicating that the presence of advanced liver fibrosis does not obviously alter the outcome.Model 4: Adenomas + advanced liver fibrosis; OR = 1.00, *p* < 0.0001, indicating that the presence of advanced liver fibrosis does not influence the outcome in any way.

## 4. Discussion

From our study, HCC emerged as both the most frequently inconclusive and misdiagnosed lesion within our cohort. Additionally, considering that the most common enhancement pattern observed in the inconclusive group was hyper-iso-iso (arterial, portal, and late phase) during CEUS, it is evident that HCC presented the most significant diagnostic challenges. The hyper-iso-iso pattern in patients with advanced fibrosis suggests a strong likelihood of HCC, according to Leoni et al. [18], displaying a high positive predictive value (PPV) and accuracy. The findings of Leoni’s study indicated that the hyper-iso pattern had a PPV of 94% and an accuracy of 77%, suggesting that this model could reliably indicate the presence of HCC, especially in primary nodules where it demonstrated even higher PPV and accuracy.

The increased rate of misdiagnosis or inconclusive results in patients with advanced fibrosis (42.8%) underscores the challenges in imaging fibrotic livers. The fibrotic tissue can distort the vascular architecture, which is crucial for the characterization of liver lesions by CEUS [19,20,21]. A potential solution can be found in the research conducted by Schellhaas et al. (2022) [22], who found that extending the observation period during the late phase significantly improves diagnostic accuracy for hepatocellular carcinoma (HCC), especially in cases of fibrotic livers. This extended phase allows for the observation of late-onset washout patterns that are often seen in well-differentiated HCCs but might be obscured by the fibrotic tissue’s altered vascular architecture. Beyond this, it is known that CT and MRI use extracellular contrast agents that may leak into the tumor interstitium, unlike CEUS microbubbles that stay intravascular. This causes all malignant liver lesions in CEUS to display washout, which may not occur in CT/MRI due to high vascular permeability [19]. In cirrhotic patients, washout indicates potential malignancy but does not always distinguish between HCC and non-HCC malignancies. Non-HCC malignant lesions typically show early (<60 s) and pronounced washout, while HCC is characterized by late (>60 s) and mild washout. Early mild or late pronounced washout suggests malignancy but is not specific to any particular type; such cases require further MRI or biopsy evaluation [23,24,25]. However, in the study of Bartolotta et al. [26], the diagnostic and prognostic performances of CEUS in cirrhotic patients were demonstrated despite the pathophysiological changes that occur in advanced fibrosis or cirrhosis. This contrasts with our findings, where a significant portion of lesions were either misdiagnosed or remained inconclusive in patients with liver cirrhosis.

To standardize and overcome some of the limitations of the method, international societies have introduced additional tools. In 2016, the American College of Radiology included CEUS in its comprehensive Liver Imaging Reporting and Data System (LI-RADS), a unique scoring system for CEUS examinations in patients with an increased risk of HCC [27]. LI-RADS provides standardized terminology and criteria for diagnosing and categorizing liver lesions, which improves the accuracy and reliability of HCC diagnosis [28]. According to recent studies [29,30,31], LI-RADS has significantly enhanced the diagnostic performance of CEUS in detecting and characterizing HCC nodules, reducing the rate of misdiagnosis and inconclusive findings.

When we categorized the misdiagnosed and inconclusive FLL based on the ultrasound (US) machines used for assessment, we observed no statistically significant differences. Specifically, the analysis involved 140 evaluations using the GE LogiqE9 and 839 using the Siemens Acuson S2000. The rates of misdiagnosis or inconclusive findings were comparable between the two groups assessed by different ultrasound machines, at 35% and 35.8% (*p* = 0.85), respectively, indicating that the type of ultrasound machine did not influence diagnostic accuracy.

Upon examining the enhancement patterns of misdiagnosed and inconclusive FLLs using CEUS, we observed the following distribution of patterns across the arterial, portal, and late phases: 61 lesions exhibited a hyper-iso-iso pattern; 39 displayed hyper-hypo-hypo; 22 presented with hyper-hyper-hyper; 21 showed hypo-hypo-hypo; and 21 lesions were indeterminate. Enhancement patterns with fewer than ten occurrences were not reported due to their infrequency. The predominance of the hyper-iso-iso pattern likely indicates challenges in accurately detecting contrast loss at the lesion level. In these instances, dynamic contrast-enhanced ultrasonography (D-CEUS) may provide valuable assistance [32]. In a recent study by Qiu et al. [33], researchers utilized dynamic contrast-enhanced ultrasonography (D-CEUS) in conjunction with the Liver Imaging Reporting and Data System Malignancy (LR-M) classification criteria. This approach was aimed at differentiating malignant liver nodules that are at high risk of developing hepatocellular carcinoma (HCC). The study assessed the effectiveness and accuracy of combining this imaging and classification methodology to enhance diagnostic confidence in identifying high-risk lesions. This could potentially enable more targeted and timely therapeutic interventions for patients at risk of HCC. Additionally, Lu et al. [34] demonstrated that synoptic reporting (SR), incorporating algorithmic diagnosis, and CEUS LI-RADS significantly improved the completeness, efficiency, and user satisfaction in the documentation and management of focal liver masses in at-risk patients compared to traditional prose reporting (PR).

Multiparametric Ultrasound (MPUS) standard ultrasound modes, such as grayscale (B-mode) and Doppler mode, are first-line imaging options in many specialties. Technological advancements in ultrasound have introduced a range of new features, increasing the versatility and effectiveness of ultrasound examinations [35]. Modern ultrasound devices now include capabilities like elastography for assessing fibrosis levels. The introduction of contrast-enhanced ultrasound (CEUS) has particularly transformed the paradigm for evaluating focal liver lesions (FLLs). CEUS allows for a detailed assessment of organ perfusion quantitatively (time-intensity curves) and through parametric imaging analysis. The development of the multiparametric ultrasound (MPUS) concept has significantly advanced the diagnostic capabilities of FLLs. This approach integrates various imaging features to provide a comprehensive evaluation of structural and functional changes within the liver. MPUS facilitates both quantitative and qualitative vascular assessments and measures tissue stiffness, enhancing differentiation between different types of liver lesions without the need for more invasive diagnostic methods [36]. Studies highlight the potential of MPUS to reduce the reliance on liver biopsies by offering precise lesion characterization. This evolution in ultrasound technology positions MPUS as a superior diagnostic tool compared to previous imaging techniques, even those traditionally deemed more advanced, particularly for liver applications [35,37,38,39].

Nonetheless, our analysis demonstrates the importance of supplementary imaging techniques such as CT, MRI, or biopsy for confirmation, particularly in cases where CEUS results are ambiguous or suggest malignancy.

A limitation of the study is that only a small subset of patients underwent core biopsy for the final diagnosis. Most lesions were diagnosed using cross-sectional imaging techniques, including contrast-enhanced computed tomography (CT) and magnetic resonance imaging (MRI). To optimize statistical analysis, we excluded lesions inconclusive on second-line imaging or with uncertain histology, which may introduce a bias by omitting real-world diagnostic challenges. Future research should include these cases for a more comprehensive analysis. Another limitation is the study design, which used a legacy database for the analysis. However, the time frame selected was based on the availability of information within this database.

## 5. Conclusions

From our cohort, 12.5% of FLLs were misdiagnosed and 11.5% were fully inconclusive during CEUS, with HCC and liver metastases being the most frequent misdiagnoses. Advanced liver fibrosis, adenomas, and atypical enhancement patterns like hyper-iso-iso were significant contributors to diagnostic inaccuracies, particularly in patients with complex lesion presentations, highlighting the need for supplementary imaging methods and standardized frameworks.

## Figures and Tables

**Figure 1 diagnostics-15-00046-f001:**
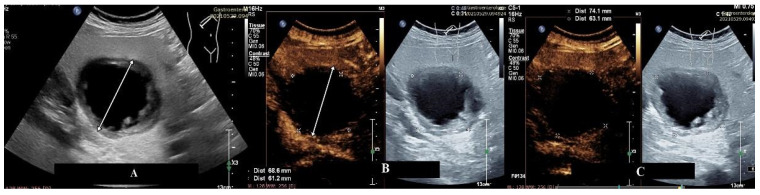
CEUS examination of an atypical enhancement pattern of liver metastases. B-mode evaluation illustrates a single lesion at the right hepatic lobe (see arrow) with an echogenic rim and anechoic center. The patient had a history of an extrahepatic malignancy diagnosed six years prior (**A**). Arterial phase at CEUS depicts a non-enhancement pattern at the rim level (**B**). The lesion maintains its non-enhancing appearance in both the portal and late phases (**C**), suggesting an avascular nature. However, subsequent contrast-enhanced CT imaging revealed the lesion to be a hypovascular liver metastasis.

**Figure 2 diagnostics-15-00046-f002:**
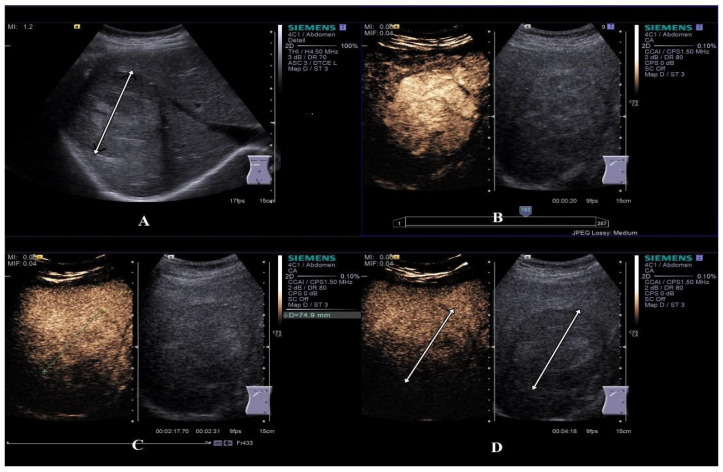
CEUS examination of a typical enhancement pattern of HCC: hypoechoic inhomogeneous liver lesion in the right hepatic lobe on B-mode ultrasonography (see arrow) (**A**). Hyper-enhancement in the arterial phase (**B)**. At the beginning of the portal venous phase, the lesion is iso-enhanced (**C**). In the late CEUS phase, the lesion is slightly hypo-enhanced suggesting washout (**D**).

**Figure 3 diagnostics-15-00046-f003:**
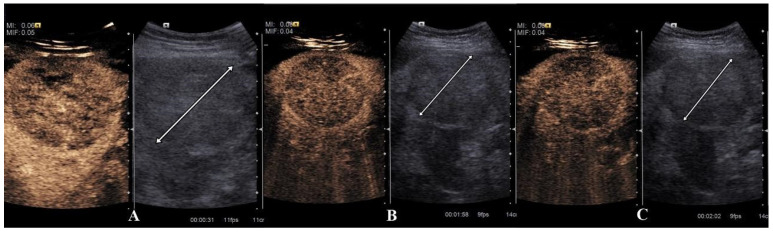
CEUS examination an atypical HCC enhancement pattern: hypoechoic in-homogeneous right liver lobe lesion on B-mode scanning (see arrow) and a hypo-enhancement pattern in the arterial phase (**A**). The hypo-enhancement pattern is maintained in the portal (**B**) and late phase (**C**). At MRI, the lesion was diagnosed as being an HCC.

**Figure 4 diagnostics-15-00046-f004:**
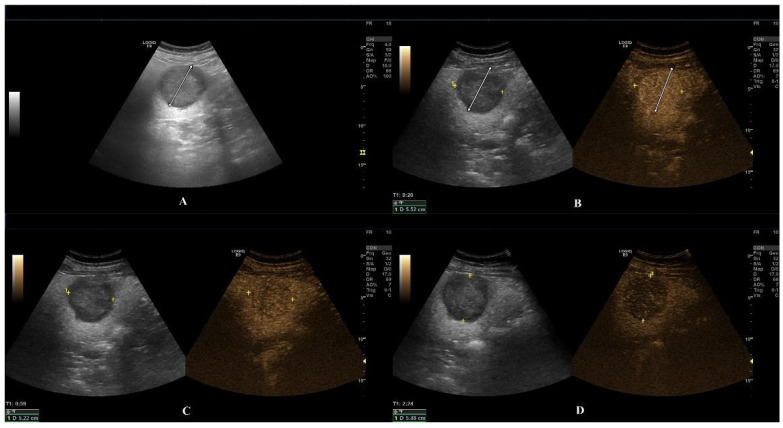
CEUS examination of a hepatic adenoma. B-mode evaluation depicts a single lesion in the right hepatic lobe characterized by a hypoechoic rim and echogenic structure, identified during a routine ultrasound examination (**A**). Arterial phase at CEUS depicts a hyper-enhancement pattern of the lesion (see markings) (**B**). In the portal and late phase, a slight washout is depicted (**C**) that becomes obvious in the late phase (see markings) (**D**). The CEUS findings initially suggested possible differential diagnoses including liver metastasis or hepatocellular carcinoma (HCC). Subsequent contrast-enhanced MRI, however, confirmed the lesion as a hepatic adenoma.

**Figure 5 diagnostics-15-00046-f005:**
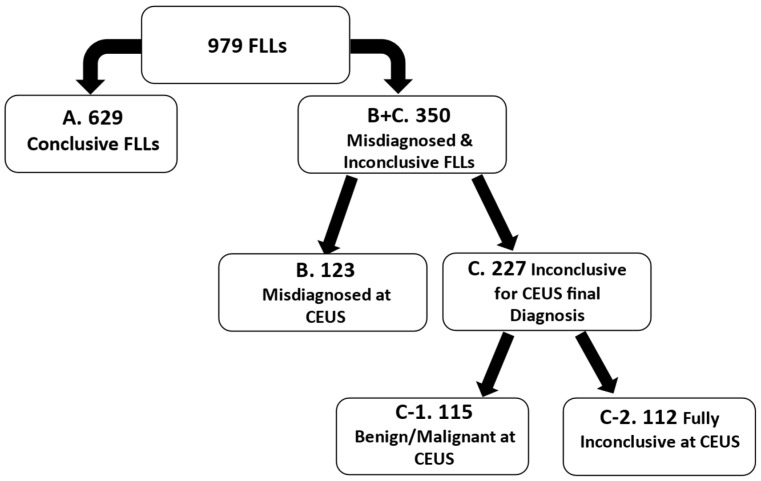
Study flowchart.

**Table 1 diagnostics-15-00046-t001:** Patients’ characteristics.

Variable	Conclusive Group	Misdiagnosed/ Inconclusive Group	*p*-Value
Age (years)	59.1 ± 13.0	60.0 ± 12.3	0.29
Gender Female Male	280 (44.0%) 349 (56.0%)	127 (36.2%) 223 (63.8%)	0.01
Known malignancy	103 (16.3%)	34 (0.9%)	<0.0001
Lesion dimension (cm)	4.5 ± 3.0	4.3 ± 3.2	0.32
Lesion characteristics Malignant Benign	369 (58.6%) 260 (41.4%)	142 (40.5%) 208 (59.5%)	<0.0001 <0.0001
CEUS diagnostic HCC Abscess Metastasis Adenoma FNH Fatty free aria Adenoma Hepatic cyst Cholangiocarcinoma Other FLLs	177 (28.1%) 40 (6.3%) 187 (29.7%) 10 (1.5%) 41 (6.5%) 18 (2.8%) 10 (1.5%) 17 (2.7%) 5 (0.7%) 124 (20.2%)	123 (35.4%) 14 (4.0%) 61 (17.4%) 14 (4.0%) 18 (5.0%) 4 (1.0%) 24 (6.8%) 4 (1.0%) 14 (4.0%) 78 (21.4%)	0.01 0.12 <0.0001 0.01 0.34 0.06 <0.0001 0.7 0.0003 0.65
Chronic liver disease	185 (29.4%)	150 (42.8%)	<0.0001

**Table 2 diagnostics-15-00046-t002:** Factors associated with misdiagnosed or inconclusive CEUS results—univariate logistic analysis.

Variable	Std Error	OR	95% CI	*p*-Value
Abscess	0.31	0.61	0.32–1.14	0.12
Adenoma	0.38	4.55	2.15–9.64	0.0001
Hepatic cyst	0.55	0.41	0.13–1.24	0.11
Other neoplasms	0.21	1.58	0.99–2.10	0.004
Fatty free	0.55	0.39	0.13–1.16	0.09
FNH	0.30	0.64	0.35–1.17	0.15
HCC	0.14	0.93	0.69–1.25	0.02
Hemangioma	0.20	0.56	0.37–0.84	0.005
Metastasis	0.16	0.47	0.34–0.66	<0.0001
Cholangiocarcinoma	0.49	5.20	1.85–14.56	0.002
Female gender	0.13	0.70	0.53–0.92	0.01
Age over 60 years	0.005	1.00	0.99–1.01	0.16
Advanced liver fibrosis	0.14	1.80	1.36–2.39	<0.0001
Lesion dimension	0.13	0.96	0.73–1.26	0.78

**Table 3 diagnostics-15-00046-t003:** Factors associated with inconclusive CEUS results—univariate logistic analysis.

Variable	Std Error	OR	95% CI	*p*-Value
Adenoma	0.38	3.90	1.81–8.40	0.0005
Other neoplasms	0.14	1.4	1.01–2.10	0.04
Hemangioma	0.22	0.60	0.38–0.93	0.06
Metastasis	0.23	0.45	0.28–0.72	0.08
Cholangiocarcinoma	0.23	5.43	1.93–14.6	0.08
Female gender	0.16	0.65	0.46–0.90	0.06
Advanced liver fibrosis	0.18	1.60	1.60–2.32	<0.0001
Abscess	0.36	0.72	0.35–1.46	0.36
Hepatic cyst	0.62	0.49	0.14–1.69	0.26
Fatty free	0.62	0.46	0.13–1.59	0.22
FNH	0.30	0.64	0.35–1.17	0.15
HCC	0.16	1.61	1.17–2.22	0.03

**Table 4 diagnostics-15-00046-t004:** Factors associated with misdiagnosed CEUS results—univariate logistic analysis.

Variable	Std Error	OR	95% CI	*p*-Value
Adenoma	0.40	11.30	5.11–25.00	<0.0001
Other neoplasms	0.23	3.02	1.01–3.78	<0.0001
Hemangioma	0.36	0.38	0.19–0.79	0.07
Metastasis	0.23	0.69	0.44–1.09	0.11
Cholangiocarcinoma	0.73	3.12	0.73–13.2	0.12
Female gender	0.19	0.93	0.63–1.38	0.74
Advanced liver fibrosis	0.22	1.16	0.75–1.80	0.48
Abscess	0.53	0.49	0.17–1.40	0.18
Hepatic cyst	1.03	0.29	0.03–2.23	0.23
Fatty free	0.09	0.27	0.03–2.10	0.21
FNH	0.39	0.99	0.45–2.18	0.99
HCC	0.34	0.22	0.11–0.44	0.0001

**Table 5 diagnostics-15-00046-t005:** Factors associated with misdiagnosed or inconclusive CEUS results—multivariate logistic analysis.

Variable	Std Error	OR	95% CI	*p*-Value
Adenoma	0.38	3.90	1.81–8.40	0.0005
Other neoplasms	0.14	1.4	1.01–2.10	0.04
Hemangioma	0.22	0.60	0.38–0.93	0.06
Metastasis	0.18	0.54	0.37–0.78	0.10
Cholangiocarcinoma	1.56	4.79	1.68–13.65	0.33
Female gender	0.14	0.77	0.58–1.02	0.07
Advanced liver fibrosis	0.16	1.39	1.00–1.93	0.04
HCC	0.23	0.96	0.87–1.01	0.001

## Data Availability

The original contributions presented in the study are included in the article. Further inquiries can be directed to the corresponding author.

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
