# Peer review of "Challenges in Diagnosing Focal Liver Lesions Using Contrast-Enhanced Ultrasound"

_diagnostics, 2024, doi:10.3390/diagnostics15010046_

Round 1

Reviewer 1 Report

Comments and Suggestions for Authors

This is a well-written article and it provides useful information for the readers who interprete CEUS examinations. I think that several minor points should be clarified before publication.

1. Figure legends of figure 4 is missing.

2. Among the reference methods, in addition to CT and MRI, a brief explanation for biopsy procedure would be beneficial.

3. Among the misdiagnosed or inconclusive lesions, can you please share the numbers of the utilized reference technique for the final diagnoses. 

4. Can you please clarify how did you assess liver fibrosis stages if all patients did not undergo liver biopsy procedure? 

5. Can you please share a brief information about the time interval between the CEUS examination and the reference technique that the final diagnosis is achieved. 

6. I think that the conclusion section should be expanded for the inconclusive lesions. In the current form it seems that the conclusion is limited for misdiagnosed lesions. 

Author Response

Dear Reviewer 1,

We sincerely appreciate your extensive work. We found your comments very helpful.

The following suggested corrections were revised as requested and were marked with track changes in the manuscript:

     1. Figure legends of figure 4 is missing

Response 1: We thank the reviewer for his/her suggestion, we added the missing legend.

  1. Among the reference methods, in addition to CT and MRI, a brief explanation for biopsy procedure would be beneficial.

Response 2: Thank you for pointing this out. We agree with this comment. Therefore, we have inserted a brief explanations in the text.

  1. Among the misdiagnosed or inconclusive lesions, can you please share the numbers of the utilized reference technique for the final diagnoses.

Response 3: We appreciate the reviewer’s suggestion; however, only 4.4% of the lesions in our cohort underwent biopsy, with the majority relying on cross-sectional imaging as the reference standard. Including detailed biopsy findings would disproportionately represent a small subset of the data and not provide additional value to the manuscript.

  1. Can you please clarify how did you assess liver fibrosis stages if all patients did not undergo liver biopsy procedure?

Response 4: Thank you for pointing this out. We agree with this comment. Therefore, we have inserted a brief explanations in the text.

“Liver fibrosis stages were assessed non-invasively using ultrasound-based techniques that can evaluate liver stiffness. The methods used in our cohort were: vibration controlled transient elastography (VCTE) performed with FibroScan® device (EchoSens, Paris, France) and 2D-SWE.GE technique offered on LOGIQ E9 from General Electric.”

  1. Can you please share a brief information about the time interval between the CEUS examination and the reference technique that the final diagnosis is achieved.

Response 5: Thank you for your comment. All reference investigations were performed during the same admission period as the CEUS examination, with the exception of core biopsies. For biopsy cases, the final histopathological results were typically available within two weeks since the princeps investigation.

  1. I think that the conclusion section should be expanded for the inconclusive lesions. In the current form it seems that the conclusion is limited for misdiagnosed lesions.

Response 6: Thank you for pointing this out. We agree with this comment. Therefore, we have inserted a brief explanations in the Conclusion section. .

Reviewer 2 Report

Comments and Suggestions for Authors

The aims of this manuscript were to retrospectively analyze the results of focal liver lesions detected by CEUS and investigate the factors leading to misdiagnosis or inconclusive results. Although the results were not novel, this manuscript still may provide useful information for clinical practices. However, the following comments should be replied before considering for publication.

1.      Methods:

The definition of inconclusive lesion in CEUS was based on all 4 ultrasound experts conclusions or only from one expert should be described.

Figure 4: no figure legends

2. Results

(a). Table 1: The item “Gender (female)” is hard to be understanded and is not a usual form for presentation. This item should be revised as Gender (male/female).

(b). From the misdiagnosed lesions 123/979, the predominant lesions were: 39 HCC; 30 metastasis; 22 hemangioma; and 11 were FNH. What are the natures of the remaining 21 lesions?

(c). Table 2: Did the item “Other neoplasms” with 95% CI 0.99-2.10 significant? The items “Hemangioma” with 95% CI 0.37-0.84 “Metastasis” with 95% CI 0.34-0.66 and “Female gender” with 95% CI 0.53-0.92 should also be described.

(d) Comparison between Table 3 and 4 showed the item “HCC” was not consistent (Table 3. Factors associated with CEUS inconclusive 95% CI 1.17-2.22, Table 4. Factors associated with CEUS misdiagnosed 95% CI 0.11-0.44) which should be described.

(e) Did the sentence ”Model 1: Advanced liver fibrosis and HCC; OR for diagnosing HCC in the presence of advanced liver fibrosis was 0.47 indicating that HCC is easier to diagnose when advanced liver fibrosis is present.” correct or require to be revised as OR for misdiagnosis HCC?

(f) The meaning of OR in Model 2 is for the diagnosis or misdiagnosis of other neoplasms or HCC?

(g) The sentence “Model 3: Advanced liver fibrosis + metastasis; OR = 0.93, p = 0.0004, indicating that the presence of advanced liver fibrosis does not significantly alter the outcome” needs to revise “not significantly” to “not obviously” because P = 0.0004 was statistically significant.

(h) Did the sentence “Model 4: Adenomas + advanced liver fibrosis; OR = 1.00, p < 0.0001, indicating that the presence of advanced liver fibrosis does not affect the outcome” correct?

3 Discussion

The sentence (line 261-263) “Non-HCC lesions typically show early (60 seconds) and pronounced washout, while HCC is characterized by late (>60 seconds) and mild washout.” was not correct.

Author Response

Dear Reviewer #2,

We sincerely appreciate your extensive work. We found your comments very helpful.

The following suggested corrections were revised as requested and were marked with track changes in the manuscript:

Introduction

Comment #1:

for his/her observation, and we corrected the manuscript.

Comment #2:

Comment #3: Table 1: The item “Gender (female)” is hard to be understanded and is not a usual form for presentation. This item should be revised as Gender (male/female).

Response: We thank the reviewer for his/her suggestion, we changed to be more understandable.

Comment #4: From the misdiagnosed lesions 123/979, the predominant lesions were: 39 HCC; 30 metastases; 22 hemangioma; and 11 were FNH. What are the natures of the remaining 21 lesions?

Response: We thank the reviewer for the question he/she raised.

3 were cholangiocarcinoma, 5 were adenomas, 4 were hepatic cysts, 4 fatty free area, and 5 were abscesses. We did not include all, because it may be hard to read.

Comment #5: Table 2: Did the item “Other neoplasms” with 95% CI 0.99-2.10 significant? The items “Hemangioma” with 95% CI 0.37-0.84 “Metastasis” with 95% CI 0.34-0.66 and “Female gender” with 95% CI 0.53-0.92 should also be described.

Response: We thank the reviewer for the question he/she raised. Yes, the result is significant as the p-value shows. We added the information in the manuscript.

 Comment #6: Comparison between Table 3 and 4 showed the item “HCC” was not consistent (Table 3. Factors associated with CEUS inconclusive 95% CI 1.17-2.22, Table 4. Factors associated with CEUS misdiagnosed 95% CI 0.11-0.44) which should be described

Comment #7: Did the sentence ”Model 1: Advanced liver fibrosis and HCC; OR for diagnosing HCC in the presence of advanced liver fibrosis was 0.47 indicating that HCC is easier to diagnose when advanced liver fibrosis is present.” correct or require to be revised as OR for misdiagnosis HCC?

Comment #8: The meaning of OR in Model 2 is for the diagnosis or misdiagnosis of other neoplasms or HCC?

Comment #11: Did the sentence “Model 4: Adenomas + advanced liver fibrosis; OR = 1.00, p < 0.0001, indicating that the presence of advanced liver fibrosis does not affect the outcome” correct?

Response: We thank the reviewer for his/her observation. Yes, the sentence is correct, but we reformulate it.

Comment #11:

Thank you for pointing this out. We agree with this comment. Therefore, we have made the changes accordingly in the text.